# Femtosecond Laser Percussion Drilling of Silicon Using Repetitive Single Pulse, MHz-, and GHz-Burst Regimes

**DOI:** 10.3390/mi15050632

**Published:** 2024-05-09

**Authors:** Pierre Balage, Manon Lafargue, Théo Guilberteau, Guillaume Bonamis, Clemens Hönninger, John Lopez, Inka Manek-Hönninger

**Affiliations:** 1Université de Bordeaux-CNRS-CEA, CELIA UMR 5107, 33405 Talence, France; 2AMPLITUDE, Cité de la Photonique, 33600 Pessac, France; 3ALPhANOV, Rue François Mitterrand, 33400 Talence, France

**Keywords:** ultrafast laser processing, femtosecond bursts, laser–material interaction, silicon, percussion drilling

## Abstract

In this contribution, we present novel results on top-down drilling in silicon, the most important semiconductor material, focusing specifically on the influence of the laser parameters. We compare the holes obtained with repetitive single pulses, as well as in different MHz- and GHz-burst regimes. The deepest holes were obtained in GHz-burst mode, where we achieved holes of almost 1 mm depth and 35 µm diameter, which corresponds to an aspect ratio of 27, which is higher than the ones reported so far in the literature, to the best of our knowledge. In addition, we study the influence of the energy repartition within the burst in GHz-burst mode.

## 1. Introduction

Femtosecond laser material processing in the so-called GHz-burst mode has attracted much attention in the last few years. The first investigations of this laser–matter interaction regime were mainly focused on metal and semiconductor ablation and have shown a significant increase in the removal rate compared to the single pulse regime [1,2,3,4,5,6]. Other studies show a lower ablation efficiency for GHz-bursts than for multiple single pulses depending on the burst parameters, such as laser fluence and number of pulses per burst [7,8,9]. Recent studies on dielectrics have added data regarding this new regime and all of them tend towards a tremendous potential for laser processing in GHz-burst mode [10,11,12,13].

Silicon lies at the heart of modern electronics, powering a vast array of devices from microprocessors to solar cells [14,15]. The advancement of silicon processing techniques has been a key factor in pushing the boundaries of device performance and functionality. Among the innovative methods employed in silicon processing, laser technology stands out as a powerful tool offering precision, speed, and versatility [16]. Laser-based silicon processing has revolutionized various aspects of semiconductor fabrication, enabling precise material modification [17,18,19], structuring [20,21,22,23,24,25], surface functionalization [26], and characterization at the micro- and nanoscale levels [27]. From doping and annealing to etching and ablation [20,28] or stealth dicing [29,30,31], lasers provide a versatile means to tailor silicon properties and structures with unprecedented control and efficiency. Different strategies were also investigated with adapted wavelengths for silicon processing, such as Bessel beams for cutting and in-volume modifications [32,33,34] or even welding [35,36].

Laser processing of silicon with GHz-bursts of femtosecond pulses has already been investigated by several groups, including ours. However, no consensus has been clearly established in the community regarding the benefit of the GHz-burst regime for silicon processing. Indeed, several publications have shown that the GHz-burst turned out to be detrimental regarding the ablation rate [37,38,39]. On the other hand, some studies show a significant increase in the removal rate [3,4,5]. Based on this divergence, we identified the need for a clear and clean comparative study on silicon processing with a fixed energy, a fixed spot size, and a fixed repetition rate.

In this contribution, we report on top-down percussion drilling and the influence of the laser parameters and temporal beam shaping on the hole depth and geometry. We led a comparative study of three different regimes, namely standard repetitive single pulses, and the MHz- and GHz-burst modes. Additionally, we were able to investigate the influence of the burst shape (i.e., the energy repartition over the pulses within the burst) in GHz-burst mode as well as the impact of the burst duration. All the results are discussed in terms of hole depth and hole morphology.

## 2. Materials and Methods

Our experiments were carried out using an industrial laser system, a modified Tangor 100 from Amplitude emitting 500 fs laser pulses at a wavelength of 1030 nm, which is described in detail in reference [40]. This flexible laser system allows for a precise optimization of the drilling parameters such as burst repetition rate, burst energy, number of pulses per burst, pulse duration, or burst shape without any change in the optical path from the laser source to the target material and the beam parameters. In this study, we investigated the influence of the laser parameters on the drilling of silicon using three different regimes (repetitive single pulse, MHz-burst, and GHz-burst), as depicted in Figure 1.

In repetitive single pulse mode, the whole energy is contained in a single pulse. For the MHz-burst mode, the same energy is divided into a burst of 2 to 32 pulses at a 40 MHz intra-burst repetition rate, and the same is true for the GHz-burst regime, with bursts of 50 to 400 pulses at a 1.28 GHz intra-burst repetition rate. The pulse and burst repetition rates were fixed at 1 kHz for the whole study in order to keep comparable conditions for the different regimes. In burst mode, and especially in GHz-burst mode, the individual pulse energy is significantly lower compared to the repetitive single pulse regime. Our laser system allows us to switch directly from one regime to another without the need to realign the experimental setup. Moreover, we have the ability to investigate the burst shape as was presented in [41] and as depicted in Figure 2 with the schematic representations of different energy repartitions we investigated in this study (a, c, e) along with the corresponding photodiode measurements acquired with an oscilloscope (photodiode EOT-3500 and oscilloscope MSO70404C, from Tektronix).

From a practical point of view, the gain depletion in the laser amplifier leads to an uneven intensity profile during the burst. This phenomenon can be pre-compensated by applying a tuned pulse energy distribution thanks to an acousto-optic modulator driven by an arbitrary wave generator (AWG) on the burst before amplification to obtain the desired burst shape after the amplification [41]. This technique allowed us to design three different burst shapes, as depicted in Figure 2: the decreasing burst shape (a,b), the flat burst shape (c,d), and, finally, the increasing burst shape (e,f). As can be observed in the measured photodiode signals in Figure 2b,d,f, the rising and falling edges of the bursts are not straight due to the rise and fall time of the acousto-optic modulator used for burst shaping. Therefore, the burst shaping process requires a certain number of pulses to be efficient. In our case, the burst shape influence was investigated with bursts of 100 pulses.

The holes were drilled by focusing a Gaussian beam on the surface of the silicon samples (monocrystalline, thickness 300 µm) using a microscope objective Mitutoyo Plan NIR Apo 5X, resulting in a measured spot size of 8.5 µm (1/e^2^ diameter). The spot size was measured using a homemade calibrated system with an uncertainty of ±0.64 µm. Using a top view Basler CMOS camera (Basler acA1920-25mu), we can visualize through the focusing objective and accurately set the position of the laser focus at the front surface of the samples, as can be seen in Figure 3. During the drilling, we use a sideview system composed of an infrared diode emitting at 1300 nm (Thorlabs, M1300L4) for illumination and an InGaAs beam profiler from Femtoeasy (BeamPro SWIR 10.8 Laser Beam Profiler, Femtoeasy, Pessac, France) used as a camera coupled with a long-distance microscope (InfiniMax KX with MX-6 Objective) for real-time imaging. The latter is equipped with a 1300 nm bandpass filter in order to visualize directly through the samples and avoid being blinded by the processing laser wavelength. The focusing head is mounted on a Z-motorized stage (VP25X, MKS Instruments), whereas the sample is fixed on a motorized XY-monolithic stage (One-XY60, MKS Instruments). The XYZ-stages and the laser gate are controlled with DMCpro software (Direct Machining Control, Vilnius, Lithuania). The workstation has a granite base and gantry, ensuring high stability and excellent repeatability of the experiments. The depth of the holes was measured by using the sideview imaging system with ImageJ software (version 1.54g). The pictures taken with this visualization system were also used to reveal the general quality of the holes.

## 3. Results

### 3.1. Silicon Drilling with GHz-Bursts of Femtosecond Pulses

This section is entirely dedicated to GHz-burst mode percussion drilling of silicon. The aim here is to provide a comprehensive study on percussion drilling with GHz-bursts. Several parameters are investigated, such as burst fluence, burst duration, and burst shape, meaning that the energy distribution within the burst is variable and can be changed and optimized.

#### 3.1.1. Burst Fluence

We investigated the evolution of the hole depth as a function of the number of bursts applied to the sample for burst fluences ranging from 30 J/cm^2^ to nearly 200 J/cm^2^. We used bursts of 50 pulses per burst (ppb) at 1.28 GHz with a burst repetition rate of 1 kHz. The image of the resulting holes obtained with a burst fluence of 170 J/cm^2^ in a drilling time range from 20 ms to 10 s is depicted in Figure 4.

In this image, we can observe a linear increase in the depth as a function of the number of bursts applied to the sample up to drilling times of about 200 ms. Then, in a second time, a saturation occurs, as observed in dielectrics [13]. However, the morphology is quite different to that observed in glasses. In this case, the holes are less cylindrical and the overall uneven shape of the hole appears similar to that which was observed for MHz-burst drilling in glasses [11,42]. The morphology also changes from the first holes to the last holes. At the beginning of the drilling, the holes are very cylindrical up to a certain depth, where the morphology of the tip of the holes becomes much more tapered. Note that the bright halo that can be observed in the background of the image results from the 1300 nm illumination diode. The graphical representation of the hole depth as a function of the number of bursts applied to the sample for four values of fluences is represented in Figure 5 for the whole drilling range. The hole depths are measured with an accuracy of ±2.5% (2 pixels on our camera); therefore, we did not add any error bars in this figure as they would not be visible.

In this figure, we can observe the same tendency as was depicted in Figure 4, a linear increase in the depth followed by a saturation of the depth. Just as was observed for dielectrics, the depth of saturation increases with an increasing fluence. It is important to note that the literature reports on a three-stage drilling process in glasses [13]. Firstly, surface ablation occurs for a low number of bursts applied to the sample; the ablation plume escapes the crater and expands freely in the air, resulting in a low screening effect and thus a high ablation rate. Secondly, deep ablation or contained ablation occurs, with the plume of ablated matter contained in the hole. Due to interactions with the inner walls of the holes, the plume gets denser, reducing the ablation rate. Finally, the saturation occurs and the drilling process stops. The transmission of the beam towards the tip of the hole diminishes and eventually the drilling stops [13]. However, contrary to those prior observations in glasses, the drilling process in silicon shown in Figure 5 displays only two steps, while there were three for glasses. We suppose that the first stage, corresponding to a low number of bursts applied to the sample, cannot be seen with our observation system due to the shadow effect near the surface. Moreover, the absorption of the 1030 nm wavelength is linear in silicon, while it relies on non-linear absorption for glasses; thus, the drilling process occurs much faster, i.e., with fewer bursts, in silicon than in glass. These two facts could mean that the first stage of surface ablation corresponding to the very first points of the graph is simply not visible.

#### 3.1.2. Burst Duration

This section is dedicated to the impact of the burst duration, corresponding to the number of pulses per burst, on the drilling process. We investigated the impact of this parameter on the GHz-burst regime with bursts of 50, 100, 200, and 400 pulses per burst, respectively. We kept the burst fluence constant at 450 J/cm^2^; only the individual pulse energy within the burst was changed by adjusting the number of pulses per burst. The experimental protocol of this study, including drilling time, is the same as in the previous section. The holes obtained for the three burst configurations are depicted in Figure 6.

As can be seen in this figure, there seems to be an optimum value of number of pulses per burst regarding both the depth of the hole as well as the overall shape. Indeed, although with 50 pulses per bursts the holes are straight, we can notice a certain waviness at 100 pulses per burst. The graphical representation of the depth as a function of the number of bursts is depicted in Figure 7 for the whole drilling range with the above-mentioned parameters. Additionally, we depict the graph with 400 pulses per burst.

This graph confirms that there is an optimum value of burst duration for a fixed burst fluence. Indeed, in these measurements, we clearly observe that a burst containing 100 pulses per burst shows a significantly higher depth than the other configurations. This confirms that there is a compromise that needs to be found between the burst duration and the pulse energy of the individual pulses of the burst. For example, the 50 pulses per burst configuration provides twice as much energy per pulse compared to the 100 pulses per burst configuration, but the burst is probably too short to benefit from accumulation [5]. Therefore, the drilling process is less efficient at 50 ppb. On the other hand, the 400 pulses per burst configuration provides a very long burst that could enhance the heat accumulation, but the pulse energy is too low to provide an efficient drilling process.

A point worth noticing here is that, during this study, we were able to drill holes up to nearly 1 mm deep in silicon with a pretty regular shape. The diameter was measured around 35 µm, which means we were able to obtain aspect ratios as high as 27, which is already higher than what was obtained in the literature [34].

#### 3.1.3. Burst Shape

In this section, we present a study of the influence of the burst shape on the drilling process. We investigated the three burst shapes introduced previously. The goal here is to determine if more intense pulses in the beginning of the burst can enhance the drilling process or if it is better to use high energy pulses at the end of the burst when the material has already been heated [43] by a smoother beginning of the burst. The flat burst is also depicted in this study as a compromise between the two other burst shapes, as it was shown in a recent study that this burst shape produced the best results in dielectrics [41]. Just as in the previous section, we investigated the evolution of the depth as a function of the number of bursts applied. The images of the holes produced with the three burst shapes are depicted in Figure 8. In this figure, we display holes obtained with a drilling time ranging from 10 ms to 10 s. The burst fluence in this case was fixed to 250 J/cm^2^ and the burst repetition rate was kept at 1 kHz with bursts of 100 ppb.

In these images, we observe that the burst shape seems to produce a limited effect on the drilling process. Indeed, regarding the hole morphology, it appears that the three laser configurations produce tapered holes with much more irregular shapes than those obtained with 50 pulses per bursts in Figure 4. However, we can still notice a linear increase in the depth in the drilling range from 10 ms to 100 ms and then a saturation for drilling times higher than 200 ms. The graphical representation of the evolution of the depth as a function of the number of bursts applied to the sample is displayed in Figure 9.

In this figure, we can observe that the drilling dynamic is pretty similar for all these three burst shapes. Just as was observed previously, the evolution of the depth displays only two stages; the first one corresponds to a linear increase in the depth as a function of the number of bursts applied to the sample and finally a saturation of the depth when the drilling is over. However, in the case of silicon, it appears that the more efficient burst shape is the increasing burst shape, as it displays a 20% increase in the depth compared to the other configurations. This could be induced by the fact that the first pulses slowly heat the material while the higher energy pulses at the end of the burst eject more efficiently the ablated matter, probably resulting in a lower screening effect.

### 3.2. Comparison of Single Pulse, MHz-, and GHz-Burst Regimes

In this section, we compare the three operating regimes—repetitive single pulses, the MHz-burst mode, and the GHz-burst mode—regarding the drilling of silicon. Therefore, we investigated as before the evolution of the depth as a function of the number of pulses (bursts) applied to the sample in these three different regimes under comparable parameter conditions and with exactly the same optical alignment. For the repetitive single pulse regime, we used pulses at 140 µJ, which corresponds to a pulse fluence of around 200 J/cm^2^. For the two burst regimes, we used a burst energy of 140 µJ as well, which corresponds to a pulse fluence of 50 J/cm^2^ for MHz-bursts of 4 pulses, a pulse fluence of 25 J/cm^2^ for MHz-bursts of 8 pulses, and a pulse fluence of 2 J/cm^2^ for GHz-bursts of 100 pulses. The resulting holes for the single pulse regime, the MHz-burst regime with 4 and 8 pulses per burst, respectively, and the GHz-burst for 100 pulses per burst are depicted in Figure 10. For the sake of clarity, we chose to display only these four configurations, but all the laser parameters described in Section 2 were investigated (MHz-burst from 2 to 32 pulses per burst and GHz-burst from 50 to 400 pulses per burst). We show the holes obtained with a drilling time ranging from 60 ms to 10 s at a burst repetition rate of 1 kHz.

From this figure, we observe that the hole morphology is very different in the three regimes. First, the repetitive single pulse regime shows very tapered holes with a larger entrance diameter. At the beginning of the drilling, the holes are conical and show a linear increase in the depth. The saturation appears after around 7 s of drilling (i.e., 7000 pulses) after the two-stage process of depth increase, as described above. The last holes show large inlet diameters and quite irregular shapes, suggesting that the material is highly affected by the high energy of the repetitive single pulses.

Secondly, in the case of the MHz-burst regime, the evolution of the depth seems to follow the same general trend as the repetitive single pulse regime; however, it displays a three-stage behavior featuring a late saturation that appears after two distinct linear increasing stages. In this case, the morphology of the holes is very different from that obtained in the repetitive single pulse regime. The MHz-burst regime provides very thin and very regular cylindrical holes even for the deepest ones on the right side of the image. Additionally, the inlet diameters of the holes are very small, probably attesting to a less affected material. It is important to note that the configuration with 8 pulses per burst produced the best hole morphology of all the holes obtained in the MHz-burst regime. Therefore, there is an optimum energy distribution within a burst for drilling holes with optimum regularity and depth. Finally, we show, as a comparison, the GHz-burst regime that was already comprehensively investigated in the previous sections. The hole morphology is the same as that shown before for the GHz-burst regime. The holes are thin with a small inlet, and the diameter is quite similar to that obtained with MHz-bursts. However, the morphology of the holes is different from the two other regimes. The GHz-burst regime provides thin holes but with an irregular shape compared to the MHz-burst regime. However, the depth saturates much sooner than for MHz-bursts, but with a very regular depth obtained afterwards. This is certainly due to the very moderate pulse energy in the GHz-burst regime.

The graphical representation of the hole depth as a function of the number of bursts (pulses in the repetitive single pulse regime) is depicted in Figure 11. The graph displays a very different behavior from one configuration to another. The single pulse regime (black lozenges) shows a linear increase in the depth as a function of the number of pulses applied with a low slope, as can be seen in the zoom of the dashed black zone shown in the inset. This stage of linear increase is followed by a first stage of saturation of the depth before a non-linear increase in the depth occurs, finally reaching saturation.

The same tendency can be observed in the MHz-burst regime (triangles) for the three configurations (4 ppb, 8 ppb, and 16 ppb) with a much higher ablation rate at the beginning point of the graph. Additionally, the depth of final saturation is much higher, especially in the cases of 4 and 8 ppb for the MHz-burst configurations. It is important to note that the saturation was well reached with MHz-bursts, as longer drilling times than 10 s did not produce holes deeper than 750 µm. These long-time drillings are not displayed here for the sake of clarity. The GHz-burst regime (circles), on the other hand, shows a very different behavior with only two stages. The latter shows only a linear increase in the depth followed by a saturation, as was observed in the previous section. Note that the depth saturates sooner than was observed in Figure 7. This can be explained by the fact that the burst fluence is lower here. Indeed, although we have access to these three regimes, the maximum available burst energy in the MHz-burst regime corresponds to the chosen 140 µJ. Therefore, in order to keep a valid comparison of the three regimes, we chose to compare them at a constant energy of 140 µJ.

It is important to note that the behavior displayed by the repetitive single pulse and the MHz-burst regimes was directly observed during the drilling thanks to our transverse observation system. Indeed, during the drillings, we were able to directly visualize that the drilling was chopped; it displayed a phase of constant depth for some time and then the depth increased again, corresponding to the graphs depicted in Figure 11.

Finally, we show that the GHz-burst regime provides a much higher ablation rate (see inset of Figure 11) to reach a depth of several hundreds of micrometers much sooner than the other regimes but at the expense of the hole quality. Note that the MHz-burst regime shows tremendous results with a very high hole quality and allowing for even deeper drillings, but it requires a longer drilling time.

## 4. Conclusions

In this contribution, we investigated top-down percussion drilling of silicon with GHz-bursts of femtosecond pulses. We investigated the influence of different parameters such as the burst fluence, the burst duration, and the energy repartition within the burst on the depth as well as on the general morphology of the holes. We show holes up to nearly 1 mm depth in silicon with an excellent morphology.

To complete this study, we compared the GHz-burst regime with two state-of-the-art regimes, the repetitive single pulse and the MHz-burst regimes, respectively. We show that the three regimes show different behaviors regarding both the evolution of the depth as well as the morphology of the holes. Interestingly, it turns out that the MHz-burst regime with 4 and 8 pulses per burst appears as the best compromise to reach a high depth with very thin and regular holes. On the other hand, the GHz-burst provides a much higher ablation rate that can eventually reduce the processing time. Moreover, the deepest holes were drilled with the GHz-burst regime with 100 ppb and a burst fluence of 450 J/cm^2^ at a burst repetition rate of 1 kHz, resulting in holes of almost 1 mm depth and 35 µm corresponding to an aspect ratio of as high as 27. This comparative study clearly revealed that the burst regimes are perfectly appropriate for silicon processing, as we suggest that the material is less affected while still providing deep drillings.

## Figures and Tables

**Figure 1 micromachines-15-00632-f001:**
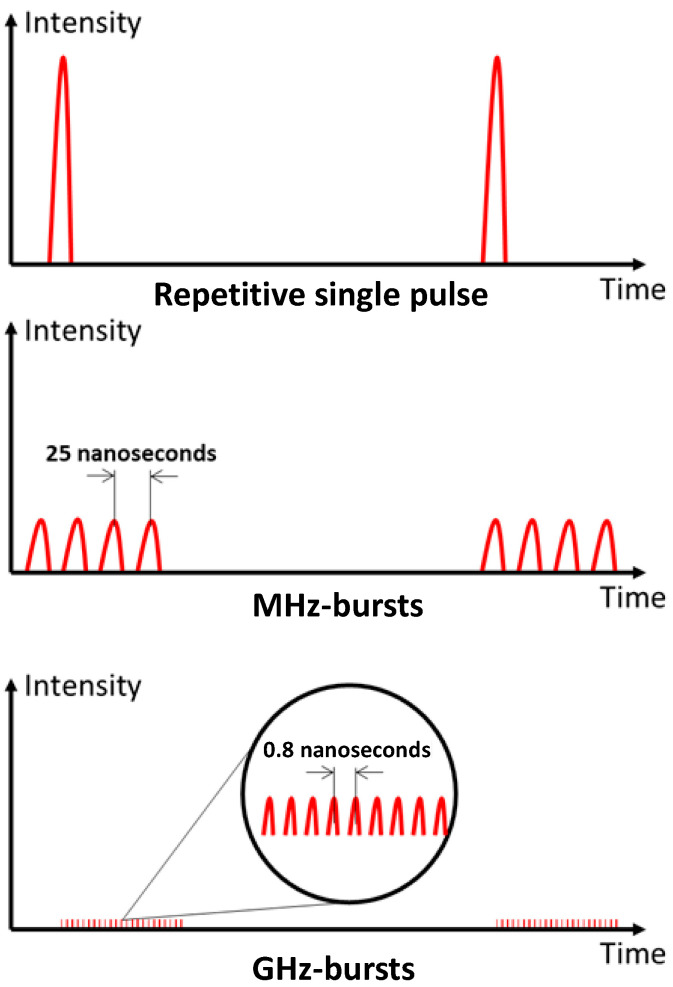
Schematic representation of the three different regimes available with our laser system.

**Figure 2 micromachines-15-00632-f002:**
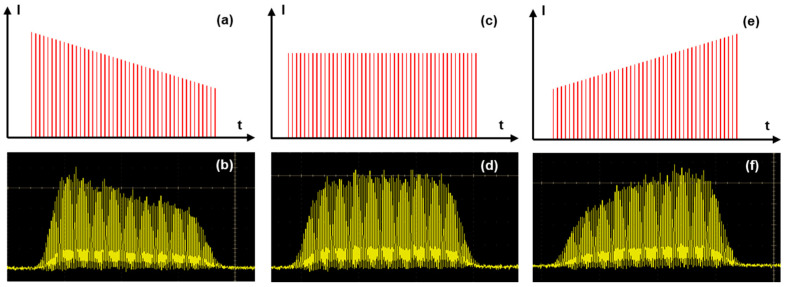
Schematic representation and measured shape of a decreasing burst (**a**,**b**), of a flat burst (**c**,**d**), and of an increasing burst (**e**,**f**).

**Figure 3 micromachines-15-00632-f003:**
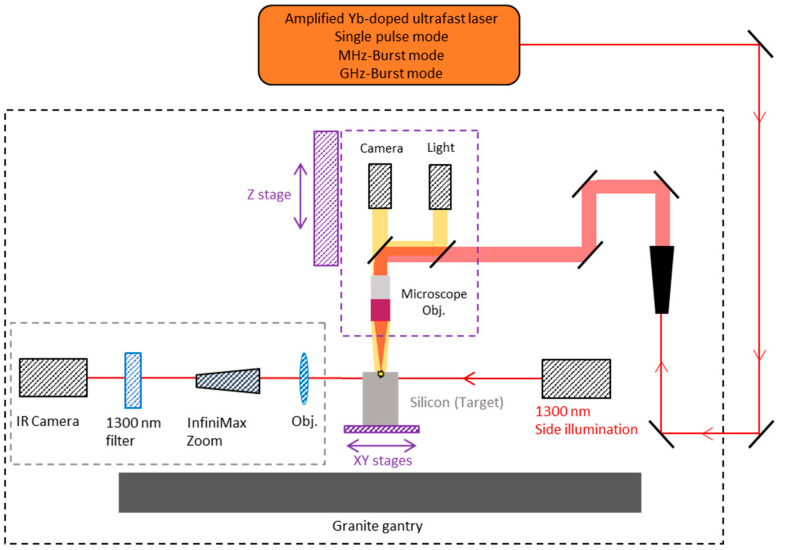
Blueprint of the experimental setup used for the drilling experiments.

**Figure 4 micromachines-15-00632-f004:**
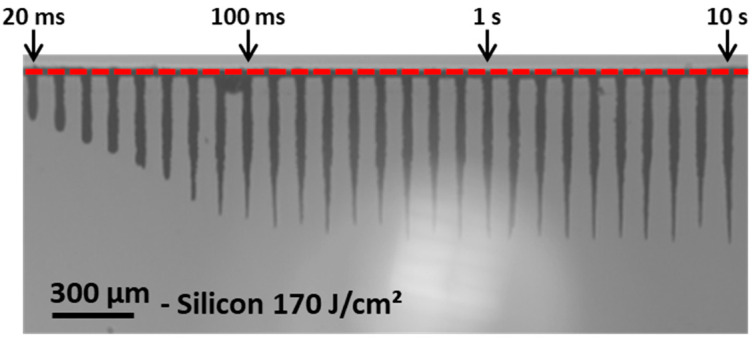
Holes obtained in silicon with a burst fluence of 170 J/cm^2^, 50 ppb, and a burst repetition rate of 1 kHz for a drilling time ranging from 20 ms to 10 s.

**Figure 5 micromachines-15-00632-f005:**
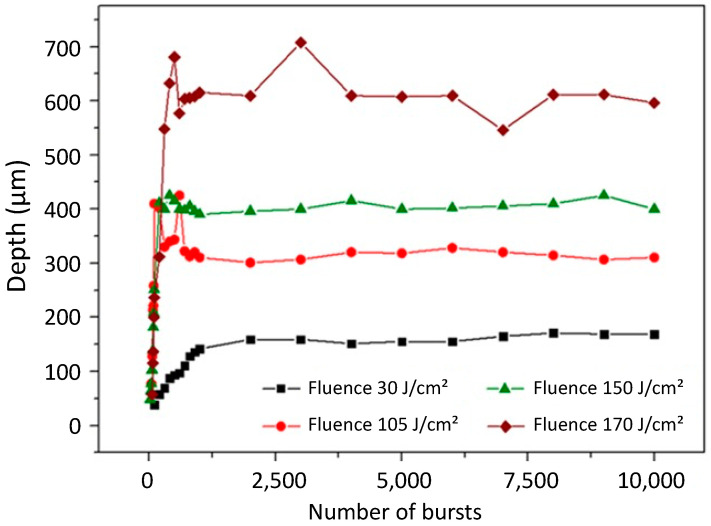
Graphical representation of the depth as a function of the number of bursts applied to the sample with a burst repetition rate of 1 kHz, 50 ppb, and a burst fluence ranging from 30 J/cm^2^ to 170 J/cm^2^.

**Figure 6 micromachines-15-00632-f006:**
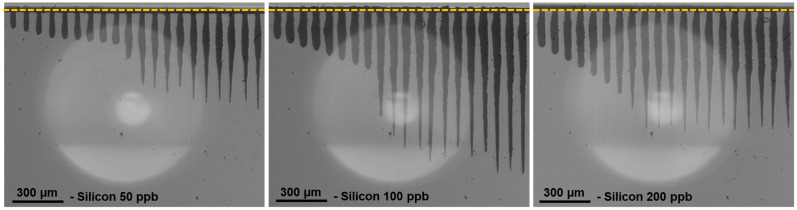
Infrared microscope images of the holes obtained in silicon for a burst fluence of 450 J/cm^2^ with a burst repetition rate of 1 kHz and a drilling time ranging from 20 ms to 3 s for 50 pulses per burst and from 30 ms to 4 s for 100 and 200 pulses per burst.

**Figure 7 micromachines-15-00632-f007:**
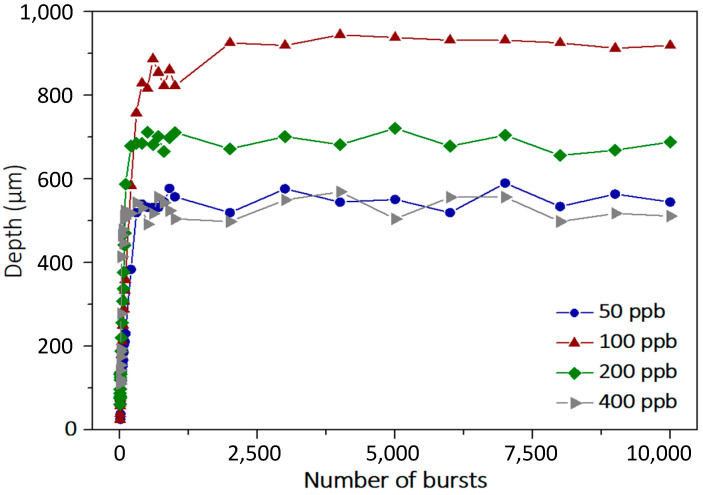
Graphical representation of the depth as a function of the number of bursts applied to the sample with a burst repetition rate of 1 kHz and a burst fluence of 450 J/cm^2^ with bursts of 50, 100, 200, and 400 pulses, respectively.

**Figure 8 micromachines-15-00632-f008:**
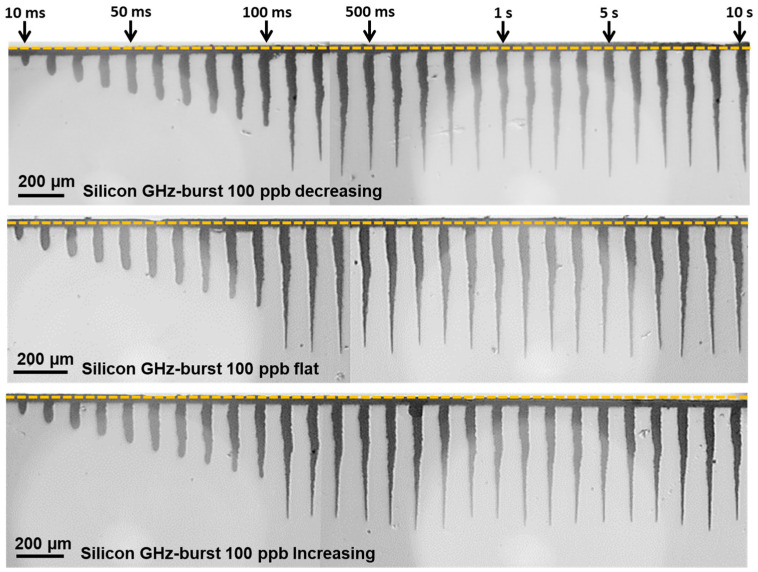
Infrared microscope images of the holes obtained in silicon with decreasing, flat, or increasing burst shape, for a burst fluence of 250 J/cm^2^ with a burst repetition rate of 1 kHz and a drilling time ranging from 10 ms to 10 s.

**Figure 9 micromachines-15-00632-f009:**
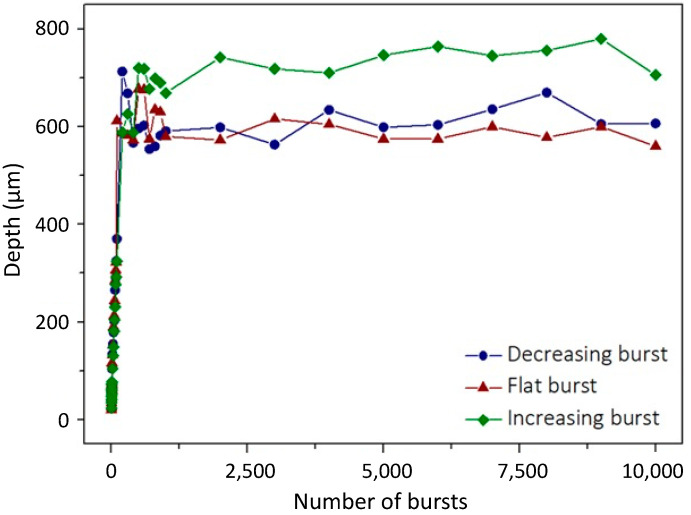
Graphical representation of the depth as a function of the number of bursts applied to the sample with a burst repetition rate of 1 kHz and a burst fluence of 250 J/cm^2^ for the three burst shapes available, each containing 100 ppb.

**Figure 10 micromachines-15-00632-f010:**
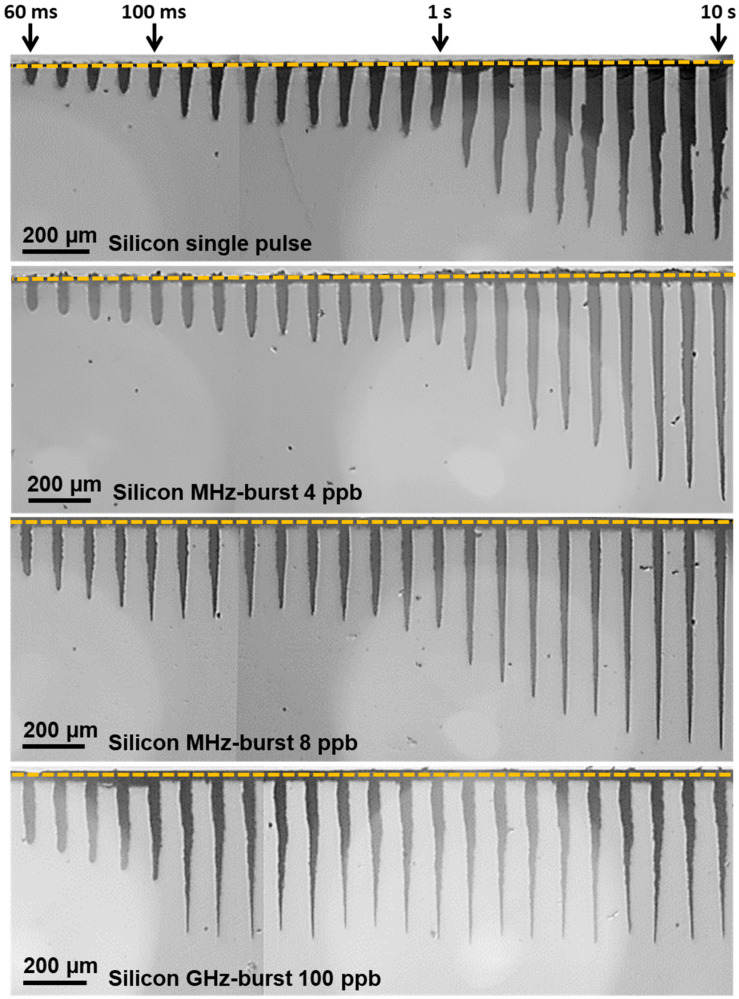
Infrared microscope images of the holes obtained in silicon for a single pulse or burst fluence of 200 J/cm^2^ with a pulse or burst repetition rate of 1 kHz and a drilling time ranging from 60 ms to 10 s for the repetitive single pulse regime, the MHz-burst regime (4 and 8 pulses per burst), and the GHz-burst regime with 100 pulses per burst.

**Figure 11 micromachines-15-00632-f011:**
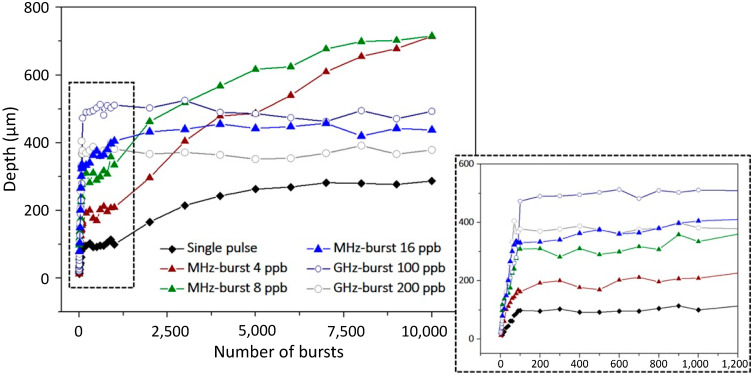
Graphical representation of the hole depth as a function of the number of bursts (pulses) applied to the sample with a burst repetition rate of 1 kHz and a burst fluence of 200 J/cm^2^ for the repetitive single pulse, the MHz-, and the GHz-burst regimes. The square in black dashed lines is a zoom of the beginning point of the graph corresponding to the first second of drilling time.

## Data Availability

Data underlying the results presented in this paper are not publicly available at this time but may be obtained from the authors upon reasonable request.

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
