# Peer review of "Femtosecond Laser Percussion Drilling of Silicon Using Repetitive Single Pulse, MHz-, and GHz-Burst Regimes"

_micromachines, 2024, doi:10.3390/mi15050632_

Round 1

Reviewer 1 Report

Comments and Suggestions for Authors

Pierre Balage et al. have present a facile result on top-down drilling in silicon, focusing specifically on the influence of the laser parameters. Moreover, the influence of the energy repartition within the burst in GHz-burst mode are also investigated. It is novel, well written and interest to the researchers in the related areas. I would consider the paper for publication after minor revisions are made according to the following specific comments:

1.      In Fig. 5, the error bar could be given.

2.      Why fs laser beam is used for drilling in silicon? Could nanosecond laser or picosecond laser achieve the same results?

3.      In Fig. 4, the entrance of the hole could be characterized or tested, such as using SEM.

4.      For the study of fs laser fabrication and it’s application, the authors may refer this recent paper: J. Cent. South Univ. (2024) 31: 1-10;

5.      For more perfection, several language mistakes could be revised.

Comments on the Quality of English Language

For more perfection, several language mistakes could be revised.

Author Response

Dear reviewer,

We would like to thank you for your careful reading and positive evaluation of our manuscript and your valuable remarks that allowed us to improve it.

Please, find here our answers:

Pierre Balage et al. have present a facile result on top-down drilling in silicon, focusing specifically on the influence of the laser parameters. Moreover, the influence of the energy repartition within the burst in GHz-burst mode are also investigated. It is novel, well written and interest to the researchers in the related areas. I would consider the paper for publication after minor revisions are made according to the following specific comments:

  1. In Fig. 5, the error bar could be given.

Our answer:

The errors bars are too small to be visible. Indeed, the uncertainty of the measurement of the hole depths is very small. Therefore, we added the following sentence in our manuscript in order to clarify (in red in the revised manuscript):

The hole depths are measured with an accuracy of ±2.5 % (2 pixels of our camera), therefore, we did not add any error bars in this figure as they would not be visible.

  1. Why fs laser beam is used for drilling in silicon? Could nanosecond laser or picosecond laser achieve the same results?

Our answer:

The use of nanosecond or picosecond pulses would create some heat affected zone, and thus, the micromachining quality of the hole would not be the same. Our decision to explore the three different regimes with femtosecond pulses on silicon is based on our recent studies of glass drilling with GHz-bursts which gave very interesting results.

  1. In Fig. 4, the entrance of the hole could be characterized or tested, such as using SEM.

Our answer:

Unfortunately, we do not have SEM images of the hole entrance. The quality of the holes looks good with our camera. We did not add any image as this does not give any additional information.

  1. For the study of fs laser fabrication and it’s application, the authors may refer this recent paper: J. Cent. South Univ. (2024) 31: 1-10;

Our answer:

We added the term "surface functionalization” in the introduction and cite the suggested paper (now reference 26).  The modified text is written in red in the revised manuscript:

“… enabling precise material modification [17-19], structuring [20-25], surface functionalization [26], and characterization at the micro- and nanoscale levels [27].”

  1. For more perfection, several language mistakes could be revised.

Our answer:

We carefully checked the manuscript and corrected some English language typos (visible in tracking mode).

With best regards on behalf of all the authors,

Pierre Balage and Inka Manek-Hönninger

Reviewer 2 Report

Comments and Suggestions for Authors

This paper present novel results on top-down drilling in silicon, the most important semiconductor material, focusing specifically on the influence of the laser parameters. In addition, was studied the influence of the energy repartition within the burst in GHz-burst mode. The article fills a gap in drilling research. All tables and figures are referenced in the text. The methodology scheme is shown and described. The cited literature is correct but requires correction (notes below). The novelty of the work must be more marked.

However, there are a several things that need to be made:

1. The abstract is too short. The abstract should contain a description of the novelties presented by the article. Moreover, the abstract lacks a description of the research results.

2. The introduction is very short. It contains a lot of literature regarding the size of the introduction. Please separate the literature and describe the cited sources in more detail. Please avoid too many multiple references.

3. The conclusions are too short, please expand the description of the research results.

 Once the changes have been made, the article may be considered for publication.

Author Response

Dear reviewer,

We would like to thank you for your careful reading and positive evaluation of our manuscript and your valuable remarks that allowed us to improve it.

Please, find here our answers:

This paper present novel results on top-down drilling in silicon, the most important semiconductor material, focusing specifically on the influence of the laser parameters. In addition, was studied the influence of the energy repartition within the burst in GHz-burst mode. The article fills a gap in drilling research. All tables and figures are referenced in the text. The methodology scheme is shown and described. The cited literature is correct but requires correction (notes below). The novelty of the work must be more marked. 

However, there are a several things that need to be made:

  1. The abstract is too short. The abstract should contain a description of the novelties presented by the article. Moreover, the abstract lacks a description of the research results.

Our answer:

This is a very good point. We have modified the abstract, and added the main result. The following sentence has been inserted (in red in the revised manuscript):

The deepest holes were obtained in GHz-burst mode, where we achieved holes of almost 1 mm depth and 35 µm diameter, which corresponds to an aspect ratio of 27 and is higher than the ones reported so far in literature, to the best of our knowledge.

  1. The introduction is very short. It contains a lot of literature regarding the size of the introduction. Please separate the literature and describe the cited sources in more detail. Please avoid too many multiple references.

 Our answer:

The manuscript is already quite long with 13 pages, and the intention is an article not a review. Therefore, we prefer not to expand the introduction.

  1. The conclusions are too short, please expand the description of the research results.

Our answer:

Thank you for the remark. Indeed, we should highlight the very high aspect ratio of the holes drilled with GHz-bursts. We added the following (in red in the revised manuscript):

“… resulting in holes of almost 1 mm depth and 35 µm corresponding to an aspect ratio of as high as 27”.

With best regards on behalf of all the authors,

Pierre Balage and Inka Manek-Hönninger